# Targeting APC/C Ubiquitin E3-Ligase Activation with Pyrimidinethylcarbamate Apcin Analogues for the Treatment of Breast Cancer

**DOI:** 10.3390/biom14111439

**Published:** 2024-11-12

**Authors:** Maria Kapanidou, Natalie L. Curtis, Sandra S. Diaz-Minguez, Sandra Agudo-Alvarez, Alfredo Rus Sanchez, Ammar Mayah, Rosette Agena, Paul Brennan, Paula Morales, Raul Benito-Arenas, Agatha Bastida, Victor M. Bolanos-Garcia

**Affiliations:** 1Department of Biological and Medical Sciences, Faculty of Health and Life Sciences, Oxford Brookes University, Oxford OX3 0BP, UK; kapanidou.m@gmail.com (M.K.); rosetteagena@gmail.com (R.A.); 2Instituto de Química Orgánica, Consejo Superior de Investigaciones Científicas (CSIC), Juan de la Cierva 3, 28006 Madrid, Spainrbenito@iqog.csic.es (R.B.-A.); 3Department of Bioingeniería, Escuela Nacional de Ciencias Biológicas (ENCB), Instituto Politécnico Nacional, Mexico City 07738, Mexico; 4Nuffield Department of Medicine (NDM), Old Road Campus, University of Oxford, Oxford OX3 7BN, UK; paul.brennan@cmd.ox.ac.uk; 5Instituto de Química Médica, Consejo Superior de Investigaciones Científicas (IQM-CSIC), Juan de la Cierva 3, 28006 Madrid, Spain; paula.morales@iqm.csic.es

**Keywords:** APC/C (Anaphase Promoting Complex/Cyclosome), apcin analogues, breast cancer, cell cycle regulation, genome instability, spindle assembly checkpoint (SAC)

## Abstract

Activation of the ubiquitin ligase APC/C by the protein Cdc20 is an essential requirement for proper cell division in higher organisms, including humans. APC/C is the ultimate effector of the Spindle Assembly Checkpoint (SAC), the signalling system that monitors the proper attachment of chromosomes to microtubules during cell division. Defects in this process result in genome instability and cancer. Interfering with APC/C substrate ubiquitylation in cancer cells delays mitotic exit, which induces cell death. Therefore, impairing APC/C function represents an opportunity for the treatment of cancer and malignancies associated with SAC dysregulation. In this study, we report a new class of pyrimidinethylcarbamate apcin analogues that interfere with APC/C activity in 2D and 3D breast cancer cells. The new pyrimidinethylcarbamate apcin analogues exhibited higher cytotoxicity than apcin in all breast cancer cell subtypes investigated, with much lower cytotoxicity observed in fibroblasts and RPE-1 cells. Further molecular rationalisation of apcin and its derivatives was conducted using molecular docking studies. These structural modifications selected from the in silico studies provide a rational basis for the development of more potent chemotypes to treat highly aggressive breast cancer and possibly other aggressive tumour types of diverse tissue origins.

## 1. Introduction

Breast cancer continues to be a leading cause of cancer death among women and the most common malignancy in women worldwide [1,2]. In Africa, Asia and South America, the incidence of breast cancer is increasing, most likely because of lifestyle changes and more robust cancer screening programmes [3]. Breasts undergo several morphological changes (e.g., hypervascularity) during pregnancy due to higher levels of oestrogen and progesterone, which induce proliferation of the ducts and lobules, respectively [4]. At the same time, the higher levels of these hormones are strong driving factors for tumour progression, especially for a pre-existing tumour where the altered microenvironment may promote it to a more aggressive type and lower survival levels.

Despite their numerous advantages, small-molecule therapies targeting the active sites of enzymes have limitations such as off-target effects on healthy cells that can lead to severe undesired side effects and rapid acquisition of drug resistance in the treated cells, thus limiting their long-term effectiveness [5]. The major obstacle that drug resistance represents can be exemplified with inhibitors of protein kinases and microtubule poisons such as taxanes (paclitaxel, docetaxel) that are used in clinical practice to treat breast cancer [6]. A more recent approach proposed to develop more effective anticancer drugs consisting of small-size inhibitors that affect specific protein—protein interactions. The anaphase-promoting complex/cyclosome (APC/C) is a large multi-subunit E3 ubiquitin ligase complex that regulates cell cycle progression and mitosis exit by promoting the polyubiquitination and subsequent degradation of specific substrates [7]. In principle, blocking mitosis progression should be an efficient strategy to induce mitotic catastrophe in cancer cells, thus suppressing cancer cell expansion [8,9]. Tight regulation of cell cycle progression by the APC/C requires its physical and sequential interaction with its two coactivators, Cdh1 (Cdc20 homologue 1) and Cdc20, which form the APC/C–Cdh1 and APC/C–Cdc20 complexes, respectively [10]. Essentially, the APC/C–Cdh1 complex is primarily active during the end of mitotic exit and the early G1 phase of the cell cycle, whereas the APC/C–Cdc20 heterocomplex controls the transition from metaphase to anaphase and mitotic exit. Moreover, in response to defects in chromosome attachment to the mitotic spindle, Cdc20 acts as an inhibitor of the APC/C through its physical interaction with Mad2, BubR1 and Bub3 to form the mitotic checkpoint complex (MCC), which binds to and inhibits the APC/C. This interaction prevents the ubiquitylation of APC/C substrates such as Securin and Cyclin B1, causing a delay in anaphase onset [11,12].

On one hand, several small-size compounds belonging to various chemical families, including BCHHD 7c and CFM-4, as well as the naturally occurring compounds GDNT, Genistein and Withaferin, have been reported to inhibit APC/C functions through interference of Cdc20 binding to the APC/C. However, none of these molecules are genuinely specific to Cdc20 [8]. On the other hand, the synthetic compound TAME (tosyl-L-arginine methyl ester) acts as a mimetic drug of the C-terminal Cdc20 region commonly referred to as the IR motif, a protein region involved in Cdc20 binding to the APC/C. TAME competes with Cdc20 (and Cdh1) to bind the APC/C, leading to inhibition of degradation of APC/C substrates [13]. The fact TAME is not cell permeable, stimulated the development of the prodrug called proTAME, a small molecule that can be processed in vivo by esterases to produce TAME inside the cell. Another inhibitor worth mentioning is NAHA, a hydroxamic acid derivative that is reported to inhibit cell proliferation and colony formation in breast cancer cells and to reduce tumour weight in vivo by downregulating Cdc20 expression [14]. However, the underlying molecular mechanism of action of this compound remains largely obscure, which greatly limits its potential use in clinical practice [15].

More recently, another small-size synthetic compound, apcin (2-(2-Methyl-5-nitroimidazol-1-yl)ethyl N-[2,2,2-trichloro-1-(pyrimidin-2-ylamino) ethyl]carbamate); also known as 3-(2-Methyl-5-nitroimidazol-1-yl)-N-(2,2,2-trichloro-1-phenylaminoethyl) propionamide, which in this study we refer to as APN, was reported as a specific Cdc20 inhibitor with a direct effect on APC/C catalytic activity. APN binds to a Cdc20 pocket defining a D-box degron motif on the side face of the WD40-domain [13], thus blocking substrate-induced Cdc20 loading onto the APC/C. This steric hindrance results in the inhibition of ubiquitylation of APC/C substrates [13,16] and an extends the time in late mitosis. The latter condition eventually triggers an apoptotic response and consequently cell death. Initial studies reported an IC50 value for APN that is in the micromolar range. More recent investigations focused on the structural modifications of APN, such as the incorporation of thiazole substituents, which resulted in APN derivatives of higher potency than APN [17,18,19,20].

In this contribution, we report a novel class of APN analogues to interfere with the process of cell division in cancer cells. The new compounds belong to the pyrimidinethylcarbamate chemical class. We show that these new APN analogues exhibit important antimitotic activity in breast cancer cells, as determined by a range of functional tests including cytotoxicity and clonogenic assays, as well as flow cytometry coupled to Annexin V-FITC/PI Double-Staining Fluorescence, in 2D and 3D cancer cells in culture. Moreover, we found the new pyrimidinethylcarbamates were also effective against cervical and ovarian cancer cells. This is important because ovarian cancer is the third most common cancer among females after breast and lung cancer. There is no screening test for ovarian cancer, and recurrence is high. Although there are novel surgical techniques for ovarian cancer and new drugs that increase the five-year overall survival rate, the five-year post-diagnosis survival rate is still less than 50% [21]. Cervical cancer is the fourth leading cause of cancer death in the global female population [22].

We also show that reversine, a small-size inhibitor of the SAC kinase Mps1, exhibits very high cytotoxicity to cancer and non–cancer cells, which greatly limits its potential use in a clinical setting. Taken together, our studies demonstrate that APN analogues grown from the pyrimidinethylcarbamate moiety exhibit higher cytotoxicity than APN. We argue that further iteration of drug design and chemical synthesis around the pyrimidinethylcarbamate moiety warrant further exploration to develop more potent drugs to treat highly aggressive cancer types with a poor prognosis and different tissue origins.

## 2. Materials and Methods

### 2.1. Chemical Synthesis

Reagents. The reagents, starting materials and solvents were purchased with high purity from Honeywell, Sigma-Aldrich, Fluorochem, Merck, Alfa-Aesar, TCI, Panreac or Scharlab (in all cases from their branch in Madrid, Spain). When necessary, they were purified using standard procedures. The Celite used was CELITE^®^ S from Sigma-Aldrich (Madrid, Spain).

General procedures. All reactions were conducted under an argon atmosphere unless otherwise stated. Reaction progress was monitored by thin-layer chromatography (TLC) using Merck silica gel 60 F254 plates with aluminium support, with a thickness of 0.2 mm (Madrid, Spain). Plate visualisation was carried out using UV light (254 and 366 nm), a 2% ninhydrin solution in n-butanol (with 3% glacial AcOH by volume), a 10% phosphomolybdic acid solution in EtOH (PMA) and a 0.04% bromocresol solution in EtOH (with 5% by volume of a 0.1 M aqueous NaOH solution). Purification of reaction products was performed by flash column chromatography using Merck silica gel (0.040–0.063 mm) (230–400 mesh ASTM) as the stationary phase and previously distilled solvents as the mobile phase. For purification by ion-exchange chromatography, the stationary phase used was DOWEX^®^ 50WX4 resin from Acros (Madrid, Spain). The eluent used is indicated in each case, and the proportions of solvent mixtures are given in volume/volume ratios. The 4 Å molecular sieve, consisting of Alfa-Aesar beads, was pre-activated by heating at 150 °C for 24 h and then allowed to cool under a stream of argon. Details of the strategy and steps involved in the synthesis of APN, Me-APN, Cbz-APN and Bzn-APN are described in the Appendix A.

Techniques and equipment. Monodimensional (^1^H, ^13^C, and NOE-1D) and bidimensional (HSQC, HMBC, COSY) nuclear magnetic resonance (NMR) spectra were acquired at 298 K in the specified solvents using the following equipment: VARIAN INOVA-300 (^1^H at 300 MHz and ^13^C at 75 MHz), BRUKER AVANCE III HD-400 (^1^H at 400 MHz and ^13^C at 100 MHz), JEOL JNM-ECZ400R (^1^H at 400 MHz and ^13^C at 100 MHz) and VARIAN SYSTEM-500 (^1^H at 500 MHz and ^13^C at 125 MHz). Chemical shifts (δ) are expressed in parts per million (ppm), coupling constants (J) are expressed in hertz (Hz), and the solvent residual signal was used as an internal reference: CDCl_3_ (^1^H-NMR: 7.26 ppm and ^13^C-NMR: 77.0 ppm), CD3OD (^1^H-NMR: 3.31 ppm and ^13^C-NMR: 49.0 ppm), DMSO-d6 (^1^H-NMR: 2.50 ppm and ^13^C-NMR: 39.5 ppm) and (CD3)_2_CO (^1^H-NMR: 2.05 ppm and ^13^C-NMR: 29.8 ppm).

### 2.2. Cell Lines and Treatments

HeLa (ATCC-CCL-2™), MCF-7 (ATCC^®^-HTB-22™), MDA-MB-231 (ATCC-HTB-26™), MDA-MB-468 (ATCC-HTB-132^TM^) and SKOV-3 (ATCC-HTB-77^TM^) cancer cell lines, human fibroblasts (ATCC-PCS-201–012^TM^) and RPE-1 (ATCC-CRL-4000^TM^) cells were grown in Dulbecco’s modified Eagle’s medium (DMEM) supplemented with 12% foetal bovine serum (FBS) and 1% penicillin and streptomycin at 37 °C in an atmosphere with 5% CO_2_ and 95% relative humidity. All stock solutions of the small-size compounds were prepared in dimethyl sulfoxide (DMSO). The Mps1 kinase inhibitor reversine was used as a positive cell toxicity control.

### 2.3. Cytotoxicity Assay

An MTT assay determined the effects of different compounds’ concentrations on cell proliferation. Briefly, HeLa, MCF-7, MDA-MB-231, SKOV-3, RPE-1 and fibroblast cells were seeded in 96-well plates in 100 μL of DMEM at 3x10^3^ cells/well density. After 24 h of incubation at 37 °C in an atmosphere with 5% CO_2_ and 95% relative humidity, reversine at a final concentration of 5 μM and compounds at 0.5, 1, 2, 4, 6, 8, 10, 20, 25, 30, 40 and 50 μM were tested. In all tests, the cells were incubated for 72 h. After this, 5 μL of 3-(4,5-dimethylthiazol-2-yl)-2,5-diphenyl-2H-tetrazolium bromide (MTT) solution (5 mg/mL) was added into each well, and the cells were incubated at 37° for 3 h. Upon completion of the incubation period, the medium was removed, and 100 μL of DMSO was added to each well, followed by agitation for 15 min to dissolve the formazan crystals thus formed. Then, 10 μM proTAME was used alone and in combination with selected compounds at a 25 μM concentration in HeLa and MDA-MB-231 cells. DMSO was used as a blank, and absorbance was measured at 570 nm with a Microplate Reader (PheraStar system BMG Labtech Ltd., Aylesbury, UK).

### 2.4. Clonogenic Assay

A total of 3x10^2^ HeLa, MCF-7, MDA-MB-231 and RPE-1 cells were plated in 6-well plates in the appropriate growth medium. After overnight attachment, the cells were exposed to the selected concentration of reversine and selected apcin analogue for 72 h. After incubation, the medium was removed and replaced with a drug-free medium, and the cells were allowed to grow for nine days in drug-free conditions. After six days of colony formation, the medium was removed, the cells were washed with cold PBS, and colonies were fixed using 10% cold formaldehyde for 20 min. The cells were stained with 0.5% crystal violet in 70% methanol for 15 min. Finally, the cells were washed and dried overnight, and the number of colonies was quantified using ImageJ software 1.54 (http://imagej.nih.gov/ij; access dates 21 and 24 October 2024). The results were analysed by one-way ANOVA using Graph Prism software version 10.3.1. and expressed as average values ± standard deviations by the Dunnett/Tukey test of a posteriori comparisons, where one asterisk (*) indicates a *p* value less than 0.05 (*p* ≤ 0.05), two asterisks (**) indicate a *p* value less than 0.01 (*p* ≤ 0.01), three asterisks (***) indicate a *p* value less than 0.001 (*p* ≤ 0.001) and four asterisks (****) indicate a *p* value less than 0.0001 (*p* ≤ 0.0001).

### 2.5. Apoptosis Test

The apoptosis test was performed with the Annexin V-FITC Apoptosis kit (Thermo Fisher Scientific, Waltham, MA, USA) following the manufacturer’s specifications. Briefly, 2 × 10^5^ cells were seeded in 6-well plates in growth media and exposed to the selected concentration of reversine and the tested compounds for 24 h. The cells were stained with annexin V-FITC and 10 μL of propidium iodide to be later analysed in the NovoCyte Flow Cytometer with NovoExpress 1.5.6 software (ACEA Biosciences, Santa Clara, CA, USA).

### 2.6. Cell Cycle Test

This assay was performed using the FxCycle PI/RNase staining kit. Briefly, 1.5 × 10^5^ HeLa cells were seeded in 6-well plates in growth media. After 24 h, the cells were synchronised with serum-free media and exposed the day after to the selected concentration of reversine and the tested compounds for 24 h. Following this incubation period, the cells were harvested, fixed in 4% formaldehyde for 60 min and stained with the FxCycle PI/RNase staining kit and 5 μL of Hoechst 33258 to be later analysed in the NovoCyte Flow Cytometer with NovoExpress 1.5.6 software (ACEA Biosciences, Santa Clara, CA, USA). PE-Texas Red was used, and a 610 nm/620 nm filter selected. For data analysis, FlowlogicTM Analysis Software, version 8.6, was used.

### 2.7. Multicellular Tumour Spheroid Assays

This assay was carried out according to a previous report [23] with the following modifications: 7.5 × 10^2^ MCF-7, MDA-MB-231, MDA-MB-468, SKOV-3 and HeLa cells were seeded in 96 ultra-low-adhesion well plates. After spheroid formation, the cells were exposed to the selected concentration of reversine and the tested compounds, and Sytox Green^®^ was added for 24 h. After incubation, 20 μL of Hoechst 33258 was added to each plate, and the plates were incubated for another two hours. The cells were photographed at a magnification of 4× in the Echo revolve microscope (ECHO, BICO) and analysed by ImageJ Software.

### 2.8. Statistical Analysis

All the experiments were conducted in triplicate, and the images were inspected by digital analysis using ImageJ software 1.54 (http://imagej.nih.gov/ij; access dates 21 and 24 October 2024), with calculation of the average Sytox Green fluorescence. Data were analysed by one-way ANOVA using Graph Prism software version 10.3.1. and expressed as average values ± standard deviations by the Dunnett/Tukey test of a posteriori comparisons, where one asterisk (*) indicates a *p* value less than 0.05 (*p* ≤ 0.05), two asterisks (**) indicate a *p* value less than 0.01 (*p* ≤ 0.01), three asterisks (***) indicate a *p* value less than 0.001 (*p* ≤ 0.001), and four asterisks (****) indicate a *p* value less than 0.0001 (*p* ≤ 0.0001). ST: Sytox Green.

### 2.9. Molecular Docking Studies

Receptor structure. In this study, a 3D model of the human cell division cycle 20 homologue (Cdc20) that was based on a crystal structure solved at high resolution (PDB ID: 4N14 and [13]) containing the apcin molecule within its binding crevice was used to dock potential Cdc20 inhibitors. Initial protein optimisation of the chosen Cdc20 3D structure was carried out using the Protein Preparation Wizard module integrated within the Maestro suite software 12.0. Release 2021-24 (Schrödinger, LLC, New York, NY, USA, 2022).

Ligand preparation. Low-energy three-dimensional conformations of each molecule (apcin, Me-apcin, Cbz-apcin and Bzn-apcin (from now on, these molecules are referenced in this report as APN, Me-APN, Cbz-APN and Bzn-APN, respectively) were obtained using the LigPrep module within Maestro suite software 12.0. Release 2021-24 (Schrödinger, LLC, New York, NY, USA, 2020). Epik software was employed to predict pKa values in the 7.0–7.5 pH range and to identify every chemically sensible structure using the Hammett and Taft methodology [24]. All compounds were minimised using the OPLS4 force field implemented in Maestro [25].

Molecular docking studies. Prior to applying XP docking within the Schrödinger package, a docking grid was generated using the receptor grid generator tool within Glide for ligand screening. To ensure the grid covered the WD40 substrate-recognition domain of human Cdc20, the dimensions of the grid were set at 26 Å in length along the x-, y- and z-axes and was centred on a co-crystallised ligand [13].

XP-Glide docking. XP Glide-dock modules integrated within the Schrödinger package were used for XP docking calculations [26,27]. The ligand poses that were generated were subjected to a series of hierarchical filters to assess ligand interactions with Cdc20. The OPLS4 force field was used for energy minimisation.

In silico ADME parameters. A set of physicochemical properties was calculated using QikProp integrated in Maestro. The swissADME webserver [28] was used to predict potential interactions of apcin and its derivatives with cytochromes P450.

PAINS identification. In order to avoid the presence of potential promiscuous moieties or PAINS (pan-assay interference compounds) [29,30], all the molecules were analysed using the swissADME webserver [28] to filter out molecules containing such chemical groups.

## 3. Results

### 3.1. Chemical Synthesis of APN Analogues

The preparation of the apcin derivatives APN (**1**), (**2**) and (**3**), which are shown in Figure 1, was accomplished through the synthetic sequence briefly summarised in Figure 1. Details of the synthesis and spectroscopy data of the new APN analogues are described in the Appendix A. To achieve the synthesis of these new antagonists of APC/C activation by Cdc20 harbouring a [(2,2,2-trichloroethyl)amino] pyrimidine structure, two synthetic approaches were developed, which we termed as the East–West Route and West–East Route (Figure 1). The latter is a linear synthesis strategy used for the synthesis of APN and its analogues, Cbz-APN and Me-APN. In contrast, the East–West Route adopts a divergent synthetic strategy through which it was possible to generate significant structural diversity from a common synthetic intermediate. This approach presented a significant advantage compared to the West–East Route and allowed rapid synthesis of the APN analogue Bzn-APN.

The synthesis of apcin (**1**) and its analogue Me-APN (**3**) began with the functionalisation of the hydroxyl group of metronidazole (**5**) and 3-[(*tert*-butoxycarbonyl) methylamino]propan-1-ol (**7**) to generate nitrophenylcarbonates (**10**) and (**11**), respectively, which subsequently led to the formation of the carbamate group present in compounds (**1**) and (**3**) through their reaction with 4-nitrophenyl chloroformate (**9**). Subsequent treatment with an aqueous solution of ammonia in methanol rendered the carbamates (**12**) and (**13**) (Figure 2A). For the preparation of carbamate (**14**), a different synthetic approach was followed in which the reaction of benzyl chloroformate (**6**) and an aqueous solution of ammonia in methanol led to the formation of this compound. The introduction of the trichloromethyl group was carried out using chloral hydrate and compounds (**12**), (**13**) and (**14**) in the absence of solvent, which resulted in the formation of alcohols (**15**), (**16**) and (**17**), respectively. The nucleophilic attack of the carbamate nitrogen group of compounds (**12**), (**13**) and (**14**) on either the *Re* or *Si* face of trichloroacetaldehyde resulted in the generation of both enantiomers of compounds (**15**), (**16**) and (**17**) (Figure 2A). The synthesis continued with the introduction of a pyrimidine moiety. For this purpose, it was necessary to activate the hydroxyl group of compounds (**15**), (**16**) and (**17**) using thionyl chloride in the presence of a catalytic amount of pyridine. This reaction led to the formation of alkyl chlorides (**18**), (**19**) and (**20**), which were used in the next synthetic step without purification. The reaction of these compounds with 2-aminopyrimidine in the presence of Et_3_N resulted in the formation of the desired products APN (**1**) and Cbz-APN (**2**), as well as the compound carbamate (**21**). To prepare Me-APN (**3**) through this synthetic route, an additional deprotection step was incorporated using the standard conditions described in the literature. To obtain the free amine, ion-exchange chromatography using DOWEX^®^ 50WX4 acid resin was required. Through the West–East Route, APN (**1**) was obtained with a higher yield (35%) than that reported in the literature [13]. The preparation of two APN structure analogues, Cbz-APN (**2**) and Me-APN (**3**), was also achieved with good overall yields of (58%) and (8%), respectively. This is important because the larger amounts of APN and Me-APN produced with this methodology facilitated the chemical synthesis of new APN analogues including Cbz-APN with an excellent yield. The preparation of Bzn-APN was carried out using an alternative synthetic strategy, which we have termed the East–West Route (Figure 2B). To this aim, 2-aminopyrimidine (**8**) was used as the starting material. Heating of amine (**8**) and chloral hydrate under reflux in THF led to the formation of alcohol (**22**). The use of a prochiral reagent along with the reaction conditions employed, as in the West–East Route, results in a racemic mixture. The subsequent treatment of compound (**22**) with thionyl chloride under reflux in dioxane resulted in alkyl chloride (**23**), which was used in the following step without purification. Then, 2-(aminomethyl)benzimidazole dihydrochloride was employed with an additional amount of Et_3_N to release the amine for subsequent substitution reactions. In this way, Bzn-APN was synthesised in three synthetic steps with a good overall yield of 46%. Moreover, the East–West Route should pave the way for the preparation of more complex APN analogues and an in-depth exploration of the structure–activity relationship to develop new lead compounds of higher potency as antagonists of APC/C activation by Cdc20 in breast cancer cells and other cancer types of different tissue origins.

### 3.2. APN, Me-APN, Cbz-APN and Bzn-APN Are Cytotoxic to HeLa, MCF-7, MDA-MB-231 and SKOV-3 Cancer Cells

Cell viability was first evaluated in 2D cultures of MCF-7, MDA-MB-231, HeLa and SKOV-3 cancer cells and in human fibroblasts and RPE-1 cells using the MTT assay. Among all the compounds tested, reversine, APN, Me-APN, Cbz-APN and Bzn-APN decreased the viability of MCF-7, MDA-MB-231 and HeLa cells after 72 h of treatment exposure (Figure 3). The measured cytotoxicity of APN at a 25 μM concentration in cancer cells was in good agreement with previous reports by others [13]. The more pronounced cytotoxic activity of all APN analogue compounds tested in a panel of cancer cells of different tissue origins was observed for Cbz-APN in MCF-7 cells, an oestrogen-responsive type of breast cancer of the class Luminal A [31]. Moreover, the relative cytotoxicity of this compound was higher than that of any other compound in MDA-MB-231, HeLa and SKOV-3 cells. This was also the case in fibroblasts and RPE-1 cells after 72 h of treatment, although to a much lower extent, arguing in favour of using Cbz-APN as a chemical scaffold to synthesise new Cbz-APN derivatives of enhanced therapeutic potential.

In all the cancer cell lines tested, Me-APN showed slightly higher cytotoxic activity than APN (e.g., ca. 15% in MDA-MB-231 and HeLa cells and ca. 10% in MCF-7, SKOV-3 and RPE-1 cells) based on MTT assays. Little cytotoxicity was observed in fibroblasts. The APN analogue Bzn-APN showed comparable cytotoxicity to APN in all the cancer cells tested, with the exception of HeLa cells, where Bzn-APN showed slightly higher cytotoxicity than APN. In all the cancer cell lines tested, Cbz-APN showed the highest cytotoxic activity and insignificant cytotoxicity in fibroblasts and RPE-1 cells. Interestingly, the ATP-competitive inhibitor reversine [32,33], which was used in this study as a reference to interfere with mitosis progression, was highly cytotoxic for both cancer and non-transformed cells when used at 5 μM (Figure 3). Reversine inhibits Mps1 with IC50 values of 6 nM and 2.8 nM for its kinase domain expressed in isolated and full-length versions, respectively [33]. In human RPE-1 and fibroblast cells, this small-size inhibitor caused 80% and 40% drops in cell viability, respectively. Clearly, such undesired biological activity in RPE-1 cells and fibroblasts greatly limits the potential use of reversine in clinical practice.

To assess if the cytotoxic effects of the new pyrimidinethylcarbamate APN analogues were due to inhibition of APC/C activation by Cdc20, apoptosis measurements and time-course degradation assays were carried out. As anticipated, APN and its analogues Cbz-APN and Bzn-APN induced apoptosis in HeLa cells, as determined by flow cytometry (Figure 4A). Time-course inhibition experiments coupled to Western blot analysis confirmed that interfering with APC/C activation by Cdc20 in breast cancer cells treated with APN and Cbz-APN impaired the proteolytic degradation of the genuine APC/C substrates Cyclin B and Cdc20 (Figure 4B). In agreement with previous reports [33,34], time-course inhibition of Mps1 kinase with reversine impaired a SAC response, resulting in accelerated mitosis progression and ultimately cell death (Figure 4B).

Furthermore, in order to confirm the highest cytotoxicity and apoptosis-inducing ability of Cbz-APN, clonogenic assays were conducted for this compound in a panel of cancer cells of different tissue origins. Cbz-APN tested in the 1–25 μM concentration range confirmed clear inhibition of cell proliferation at the 25 μM concentration, resulting in 42%, 84% and 75% inhibition of colonies in HeLa, MCF-7 and MDA-MB-231 cells, respectively, compared to the corresponding untreated cells Figure 5A,B). In contrast, no significant differences between the control and the different concentrations of Cbz used were noted in RPE-1 cells. This clearly indicates that Cbz at a concentration of 25 uM did not inhibit the growth of this type of normal cells. A computational image analysis enabled the quantification of clones, which together with a statistical analysis (i.e., *p* values) indicated significant differences between untreated and treated cells at increasing concentrations of Cbz-APN.

### 3.3. The Combined Use of APN, Me-APN, Cbz-APN and Bzn-APN with proTAME Results in a Synergistic Effect

Previous studies on APN have shown a synergistic cytotoxic effect when this compound was used in combination with proTAME [13]. This feature prompted us to ask whether the new APN analogues could induce a similar biological response. To this aim, Me-APN, Cbz-APN and Bzn-APN were tested in combination with proTAME. The prodrug proTAME defines a different chemical class of antagonists of APC/C activation by Cdc20. This prodrug is hydrolysed in vivo by esterases to render the active compound TAME inside the cell [16]. The molecular mechanism of action of proTAME differs from that of APN. Namely, TAME binds to the IR motif located by the C-terminal of the WD40 domain of Cdc20, thus disrupting the recruitment of this coactivator to the APC/C complex. The disruption of the Cdc20-APC/C physical interaction by TAME impedes APC/C activation and subsequent degradation of APC/C substrates by the proteasome. The different mechanisms of action of TAME and APN prompted early studies to test the potency of the former APC/C activation inhibitor when used in combination with APN. Such studies revealed that the prodrug proTAME, when used at low micromolar concentrations in combination with APN, exhibited a synergistic cytotoxic effect on cancer cells [13]. Based on such findings, in this study, APN, Me-APN, Cbz-APN and Bzn-APN were used at 25 μM alone and in combination with 10 μM proTAME and tested in MDA-MB-231 triple-negative breast cancer and HeLa cervical cancer cells (Figure 6A,B). In both cases, APN and the new pyrimidinethylcarbamate APN analogues exhibited a synergistic cytotoxic effect when used in combination with proTAME compared to the observed cytotoxicity of the individual compounds. The Cbz-APN–proTAME combination showed the highest cell proliferation inhibition. This finding is interesting and important considering that inhibition of APC/C activation by APN and proTAME seems to involve distinct molecular mechanisms and because the combination of cell cycle-targeted therapies may be key to enhance a therapeutic response in breast, cervical and ovarian cancer cells, which is an important aspect that we will address in future investigations. Even if APN is effective at a relatively large concentration (e.g., in the low millimolar range) and proTAME is a prodrug that requires activation by esterases before it reaches the target cells [16], the combined use of our novel APN analogues such as Cbz-APN with proTAME and/or derivatives of this prodrug represents an interesting therapeutic opportunity to treat diverse types of aggressive cancers. Although TAME is reported to mimic the IR tail of Cdc20 [12] and is expected to specifically interfere with the IR tail-dependent interaction of Cdc20 with the APC/C, it must be noted that cryoEM structures of the APC/C reported by the Barford group have shown that TAME can also bind to certain subunits of the APC/C. Therefore, further studies on proTAME/TAME are required to fully elucidate the molecular mechanism of action of this inhibitor before it can be considered for use in clinical practice either alone or in combination with other drugs.

### 3.4. APN, Me-APN, Cbz-APN and Bzn-APN Inhibit Cell Proliferation in 3D Cancer Cells

Finally, in order to confirm that Cbz-APN exhibited the highest cytotoxicity to Hela, MCF-7, MDA-MB-231 and MDA-MB-468 breast cancer cells and HeLa and SKOV-3 cervical and ovarian cells, respectively, but was far less cytotoxic to human fibroblasts and RPE-1 cells in 2D cells in culture, we investigated the relative activity of these compounds in 3D cancer cells. This type of cells, also known as spheroids, more closely recapitulate the features of real tumours. Spheroids of HeLa, MCF-7, MDA-MB-231, MDA-MB-468, SKOV-3 and RPE-1 cells were generated and treated with the compounds APN, Me-APN, Cbz-APN and Bzn-APN. The extent of cell lysis due to the induction of apoptosis was qualitatively estimated based on the amount of DNA released upon cell lysis. DNA was stained with the fluorescent dye SYTOX, a stain that does not cross the membranes of live cells, has a >500-fold increase in fluorescence upon nucleic acid binding and is useful to stain dead cells. The results of these studies are summarised in Figure 7A. Cbz-APN proved to be a potent cytotoxic compound against breast, cervical and ovarian spheroids as estimated from the amount of DNA released upon cell lysis and thus high green fluorescence intensity. These qualitative studies also confirmed the high cytotoxic effect of the ATP-competitive inhibitor reversine in tumour cells but also RPE-1 cells (Figure 7A). Admittedly, it is very difficult to observe the effect of individual compounds in precluding spheroidogenesis without a quantitative determination of DNA staining. We addressed this technical limitation through a digital analysis of the spheroids and constructed a graph with numerical data and statistical significance (e.g., *p* values) (Figure 7B). In brief, reversine presented a greater cytotoxic effect in HeLa and MDA-MB-468 cells, Bzn-APN showed greater cytotoxicity in MCF-7 and SKOV-3 cells, and Cbz-APN was the most cytotoxic in MDA-MB-231 cells. In relation to the RPE-1 control cell line, the cytotoxic effect caused by reversine was greater than those of all the other treatments, with Bzn-APN being the one with the least cytotoxic effect. A similar extent of cytotoxicity induced by reversine and Bzn-APN was observed in HeLa cells and by reversine and APN in MCF-7 cells. Regarding SKOV-3 cells, the cytotoxicity induced by reversine was comparable to that of cells treated with APN, Me-APN and Cbz-APN and slightly lower—but with statistical significance—than that of Bzn-APN. Last, the extent of cytotoxicity induced by Bzn-APN in MDA-MB-231 and MDA-MB-468 breast cancer cells was slightly less than that of reversine.

### 3.5. Structure-Guided Docking Analysis of the New APN Analogues

There is mounting evidence of the druggability of human Cdc20 to interfere with APC/C functions in cancer cells, where the attention has been centred on the development of compounds that bind the Cdc20 D-box located in the C-terminal of the WD40 domain. APN is an archetypic small compound that targets this Cdc20 degron, thus interfering with the physical interaction between Cdc20 and the APC/C and the subsequent substrate recognition of the later multiprotein assembly. The crystal structure of the Cdc20–APN complex revealed a hairpin conformation inside the Cdc20 substrate-recognition binding site (Figure 8A). Nitrogen atoms of the APN molecule enable the formation of hydrogen bonds with negatively charged amino acids of the WD40 domain such as Asp177, while the pyrimidine group interacts via pi–pi stacking with another key residue, Trp209, as well as the nearby Tyr207. The APN trichloromethyl group, on the other hand, is located within a hydrophobic pocket. The 3D structure of this complex guided the docking analysis of new APN analogues, which included calculation of per-residue van der Waals (VdW), the electrostatic contributions (in kcal/mol) of APN and the new APN analogues and the Glide docking score (Appendix A). We and others have shown that APN exhibits moderate cell proliferation inhibition activity in vitro. It is reassuring that this property has also been found in vivo when used as the warhead of Cdc2-targeting chimaeras [35]. Importantly, a large chemical space remains to be explored to develop more potent antagonists of APC/C function by interfering with the Cdc20–APC/C interaction. Indeed, one recent study described a series of ureido-based APN analogues, which showed that these compounds performed better than APN in terms of antiproliferation activity in MCF-7, MDA-MB-231, HepG2 and HeLa cells [18]. Our computational modelling studies of a different type of APN analogues indicated that Cbz-APN can establish three H-bonds with the amino acid residue Asp177 and pi–pi stacking interactions with residues Tyr207 and Trp209 (Figure 8B). An electrostatic contribution involving the Asp177 residue enables the establishment of three H-bonds with Cbz-APN, contributing to a higher docking score when compared to the template compound, APN. It is important to mention that although there were other compounds with higher docking scores, Cbz-APN was the compound predicted to exhibit the most favourable pharmacokinetic properties (Table 1), particularly high Caco-2 cell permeability and good oral absorption.

Regarding the new APN analogue Bzn-APN, despite displaying the most compact conformation within the binding site of Cdc20, our computational docking studies indicate that this compound is able to establish H-bonds with the human Cdc20 residue Asp177 and to interact with residues Tyr207 and Trp209 via pi–pi stacking interactions (Figure 8C). As shown in Appendix A, the electrostatic contribution of Asp177 for binding this compound (−4.47 kcal/mol) considerably surpassed those of the other two new APN analogues Me-APN and Cbz-APN, as well as APN, indicating an important contribution of the amino acid residue Asp177 to the overall docking score of Bzn-APN. Me-APN, on the other hand, was predicted to have the most extended conformation within the binding site among all the studied compounds (Figure 8D).

The lack of a second aromatic group seems to facilitate its stretched-out positioning, preventing the formation of stabilising interactions with other residues beyond Asp177 and Trp209. Despite this feature, Me-APN rendered the highest Glide score among all four compounds, surpassing the score of the template compound, APN. This unanticipated score may be due to a significant electrostatic contribution of Asp177 and other residues located in the vicinity of the methyl group, such as Glu180, to stabilise the protein-inhibitor complex (Appendix A). However, the improved Glide score did not correlate with a more favourable pharmacological profile since the physicochemical descriptors for this compound were similar to those of APN and less favourable than those of Bzn-APN and Cbz-APN (Table 1).

This finding led us to infer that a second aromatic group besides the one in the unaltered core on APN might be a desirable feature to consider for the design of novel binders of the Cdc20 D-box degron with more potent antiproliferation activity in cancer cells.

It has been estimated that 50 to 90% of drug-like compounds are substrates of cytochromes P450 (CYPs), particularly of the CYP1A2, CYP2C19, CYP2C9, CYP2D6 and CYP3A4 isoforms [28,36]. This is important because inhibition of these isoenzymes is an important factor of pharmacokinetics-related drug–drug interactions, leading to toxic or other undesired adverse effects. In an attempt to understand the plausible reason(s) of the varying extent of cancer cell cytotoxicity induced by APN and its analogues in more detail, an in silico pharmacokinetics analysis was carried out. Our studies predicted Bzn-APN and Cbz-APN as substrates/inhibitors of the majority of CYP isoforms (Table 2).

## 4. Discussion

All the new types of pyrimidinethylcarbamate APN analogues—Me-APN, Bzn-APN and Cbz-APN—exhibited moderate cytotoxic activity against cancer cells in monolayers, with the latter compound showing the most pronounced effect. Taken together, our studies show that the benzyl ring favours the binding to the D-box degron pocket of Cdc20 compared to the nitroimidazole of APN. In contrast, Me-APN, which harbours a short aliphatic chain, showed cytotoxicity comparable to that of APN in all the cancer cell lines tested, whereas bulkier ring systems such as the benzimidazole moiety of Bzn-APN seems to destabilise the interaction with the target protein. These findings warrant further exploration of the chemical space using APN benzyl ring derivatives as well as different aromatic ring systems. In this regard, future work will aim to further explore the West–East synthesis route to develop Cbz-APN derivatives that harbour these chemical structure features.

APN, Me-APN, Cbz-APN and Bzn-APN all showed a synergistic cytotoxic effect when used in combination with proTAME, with the Cbz-APN–proTAME combination showing the highest synergistic effect against TNBC and cervical cancer cells. Hence, Cbz-APN has potential for its use in combination with other drugs, such as microtubule poisons, and/or new compounds structurally related to TAME/proTAME. We were surprised by the striking similarity of the combined effect of the new APN analogue with proTAME in HeLa and MDA-MB-468 cells. Future work will aim to establish thorough dose–response curves for these and other cancer cell lines treated with the new pyrimidinethylcarbamate apcin analogues and proTAME.

Our studies in spheroids showed that Cbz-APN exhibited higher cytotoxicity than APN in 3D MDA-MB-231, MDA-MB-468 and HeLa cells, which was in good agreement with the cytotoxicity data from MTT assays in 2D cells (Figure 7 and Figure 3, respectively). This is important because spheroids represent an appropriate in vitro model of cancer to study the extent of drug penetration in treated cells. The exception was MCF-7 and SKOV-3 spheroids, where the cytotoxicity of Bzn-APN appeared higher than that of Cbz-APN in 3D cells but not in 2D cells (Figure 7 and Figure 3, respectively). Future work will aim for a more in-depth analysis of Bzn-APN and new chemical structure-related derivatives in MCF-7 and SKOV-3 spheroids. We found the cytotoxic activity of the new APN analogues encouraging because they clearly achieved good penetration in 3D cells, which is an important pharmacological limitation found in diverse cancer drugs currently used in clinical practice. Mps1 is a substrate of the APC/C, and inhibition of Mps1 kinase with reversine is known to impair SAC activation and mitosis progression, resulting in cell death [32,33,34,37]. Our studies confirmed this notion as a high percentage of cell death was noticeable in 2D and 3D MCF-7, MDA-MB-231, HeLa and SKOV-3 cells treated with reversine (Figure 3 and Figure 7, respectively). Computational docking analyses of the new APN analogues provided important clues about the molecular interactions underpinning their mode of binding to Cdc20 and assisted the exploration of the chemical space to identify potential new druggable sites. Furthermore, an in silico pharmacokinetics analysis predicted the interactions of Cbz-APN with the CYP isoenzymes CYP1A2, CYP2C19, CYP2C9 and CYP3A4. Given the prominent role of CYPs in drug pharmacokinetics, future work should aim to experimentally test this prediction.

Taken together, the biological assays described in this study show that APN and its pyrimidinethylcarbamate analogues Me-APN, Cbz-APN and Bzn-APN all exhibited moderate cytotoxicity to all the cancer cell types tested in this study (breast, cervical and ovarian) when used in the low micromolar concentration range as determined using a range of functional assays. Cbz-APN showed the highest cytotoxicity against TNBC and hormone-responsive breast cancer, as well as cervical and ovarian cancer cells, in both 2D and 3D cells in culture. This new pyrimidinethylcarbamate analogue induced apoptosis in the treated cells, and this was due to the impairment of proteolytic degradation of the APC/C substrates Cyclin B and Cdc20 by the proteasome. Therefore, this new pyrimidinethylcarbamate analogue effectively interfered with APC/C substrate recognition, a direct consequence of the lack of activation of this E3 ubiquitin ligase multiprotein complex by Cdc20.

## 5. Conclusions

The pyrimidinethylcarbamate APN analogues Me-APN, Cbz-APN and Bzn-APN showed cytotoxic activity against breast cancer cells. Cbz-APN showed the highest antiproliferation activity in all the cancer cell types tested and can be used as chemical scaffold for designing new APN analogues of higher potency. Blending cell cycle-targeted strategies like the one described in this study with other cancer treatment approaches such as drugs that target different cell signalling or metabolic pathways—alone or in combination with radiation treatment—can lead to a more effective therapy to treat breast cancer. Due to the molecular mechanism of action, the use of pyrimidinethylcarbamate APN analogues may be applicable to treat other cancer types with a poor prognosis that constitute important yet unmet medical needs.

## Figures and Tables

**Figure 1 biomolecules-14-01439-f001:**
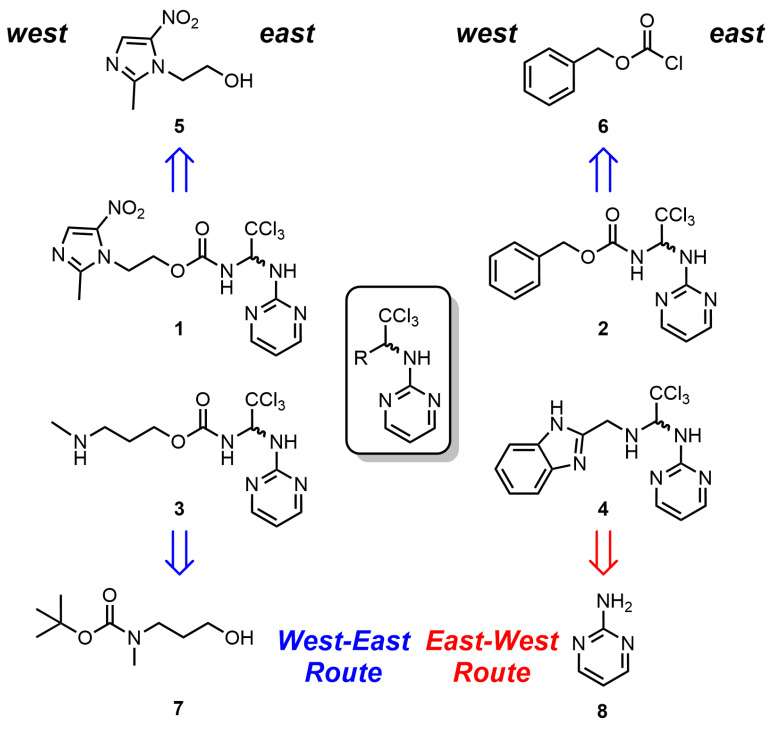
Chemical synthesis strategy of APN analogues. The simplified representation shows the route used for the synthesis of the compounds apcin (APN) (**1**); Cbz-APN (**2**); Me-APN (**3**) and Bzn-APN (**4**). The compounds **5**, **6** and **8** were purchased from commercial suppliers Alfa-Aesar and Sigma-Aldrich. The compound **7** was prepared following standard amino group protection procedures from 3-(methylamino)propan-1-ol and obtained from the commercial supplier TCI.

**Figure 2 biomolecules-14-01439-f002:**
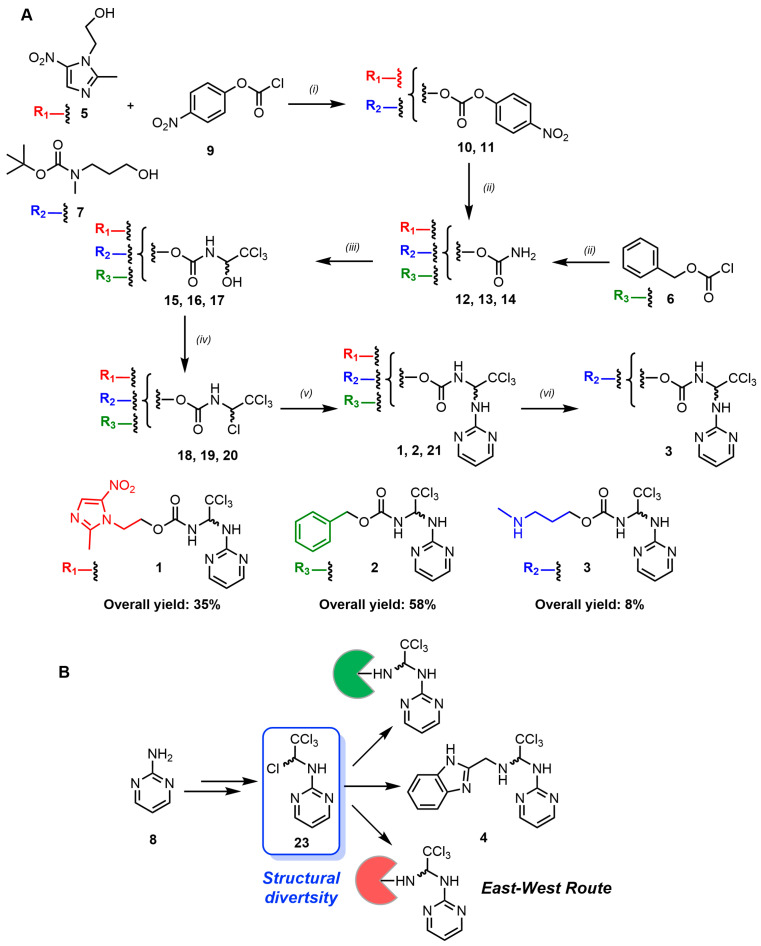
Brief description of the synthesis and yield of the new APN analogues (**1**), (**2**) and (**3**). (**A**) The steps involved were as follows: *(i)* Et_3_N, CH_2_Cl_2_, 0 °C → r.t.; *(ii)* NH_4_OH, MeOH, 0 °C → 10 °C; *(iii)* Chloral hydrate, r.t. → 100 °C; *(iv)* SOCl_2_, Py, r.t. → reflux; *(v)* 2-aminopyrimidine, Et_3_N, CH_3_CN, r.t. → reflux, *(vi)* CF_3_COOH, CH_2_Cl_2_, r.t. (**B**) The East–West Route of synthesis enabled the generation of a wide range of structurally diverse compounds.

**Figure 3 biomolecules-14-01439-f003:**
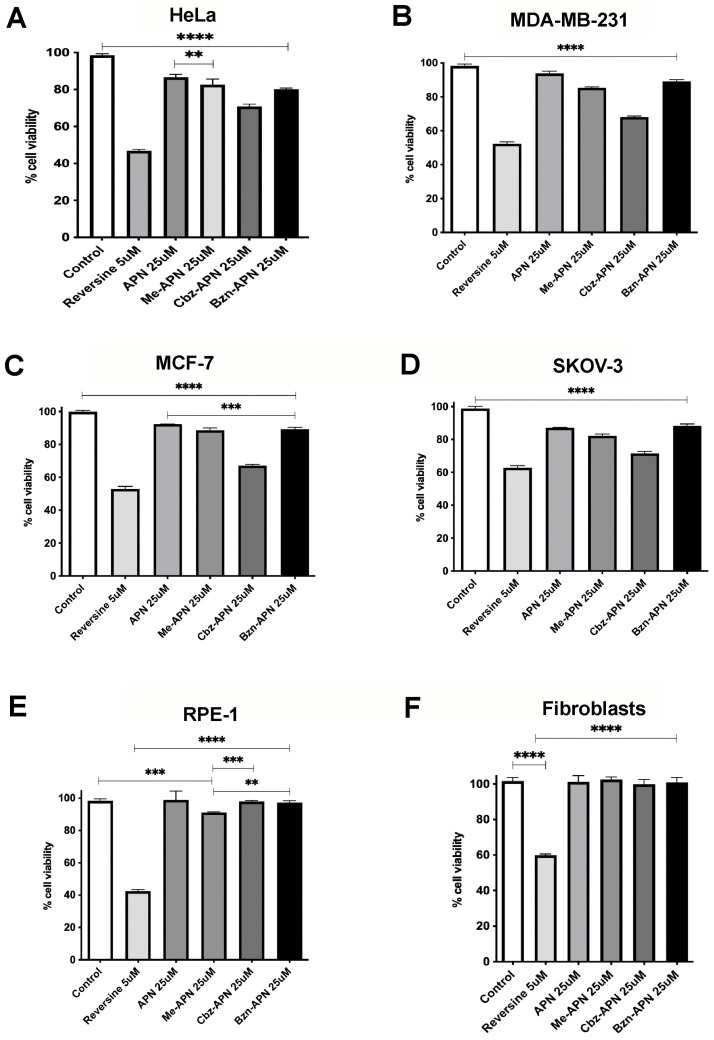
Antimitotic activity of APN and its analogues Me-APN, Cbz-APN and Bzn-APN. (**A**) HeLa cells; (**B**) MDA-MB-231 cells; (**C**) MCF-7 cells; (**D**) SKOV-3 cells; (**E**) RPE-1 cells; (**F**) Fibroblast cells. The bars represent the mean value of three independent experiments, and the lines represent the standard deviation. Two asterisks (**) indicate a *p* value less than 0.01 (*p* ≤ 0.01). Three asterisks (***) indicate a *p* value less than 0.001 (*p* ≤ 0.001). Four asterisks (****) indicate a *p* value less than 0.0001 (*p* ≤ 0.0001). Details about the number of cells seeded, temperature, incubation times with the compounds, etc., for these cytotoxicity assays are described in the Materials and Methods.

**Figure 4 biomolecules-14-01439-f004:**
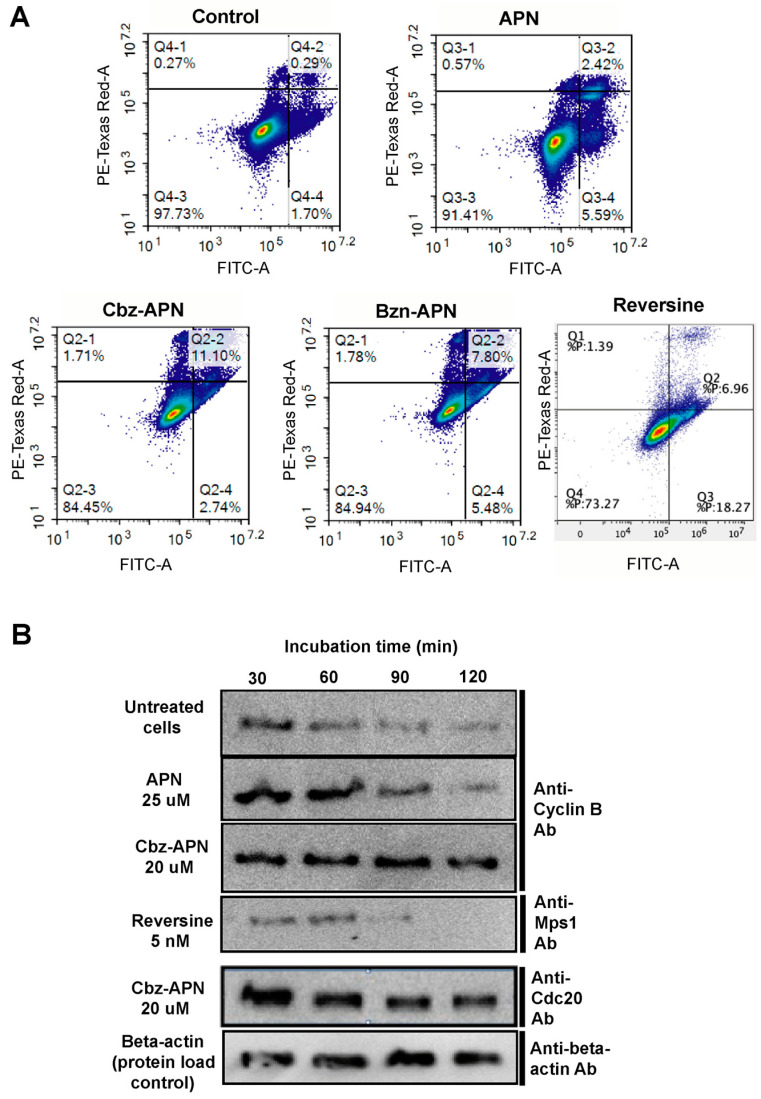
The new APN analogues induced apoptosis and APC/C substrate degradation. (**A**) Flow cytometry analysis of HeLa cells treated with APN, Cbz-APN and Bzn-APN. (**B**) Example of a time-course experiment indicating impaired proteolytic degradation of the APC/C substrates Cyclin B and Cdc20 in MDA-MB-231 cells independently treated with APN and Cbz-APN (25 and 20 µM concentrations, respectively). Cell samples were harvested at the indicated time points. Original images can be found in Appendix A.

**Figure 5 biomolecules-14-01439-f005:**
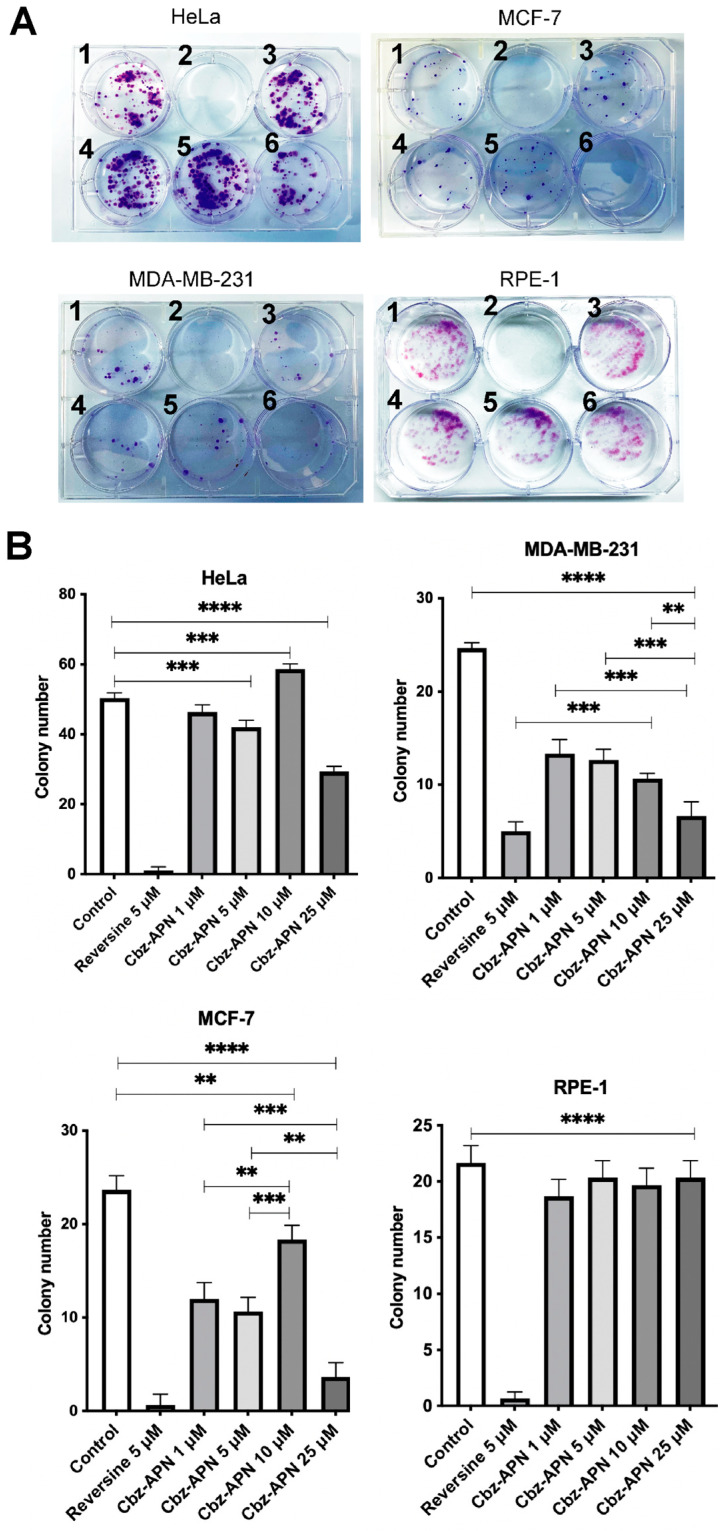
Clonogenic assays of Cbz-APN. (**A**) Well 1 = Untreated cells; well 2 = reversine at 5 μM; wells 3, 4, 5 and 6, = Cbz-APN at 1, 5, 10 μM and 25 μM, respectively. Three independent experiments were carried out for each condition. A similar clone number was consistently observed in all the replicas. Details about the number of cells seeded, temperature, incubation times with the compounds, etc., for these clonogenic assays are described in the Materials and Methods. (**B**) Quantification of the number of colonies. The results were analysed by one-way ANOVA and expressed as average values ± standard deviation by Dunnett/Tukey test. Two asterisks (**) indicate a *p* value less than 0.01 (*p* ≤ 0.01), three asterisks (***) indicate a *p* value less than 0.001 (*p* ≤ 0.001), and four asterisks (****) indicate a *p* value less than 0.0001 (*p* ≤ 0.0001).

**Figure 6 biomolecules-14-01439-f006:**
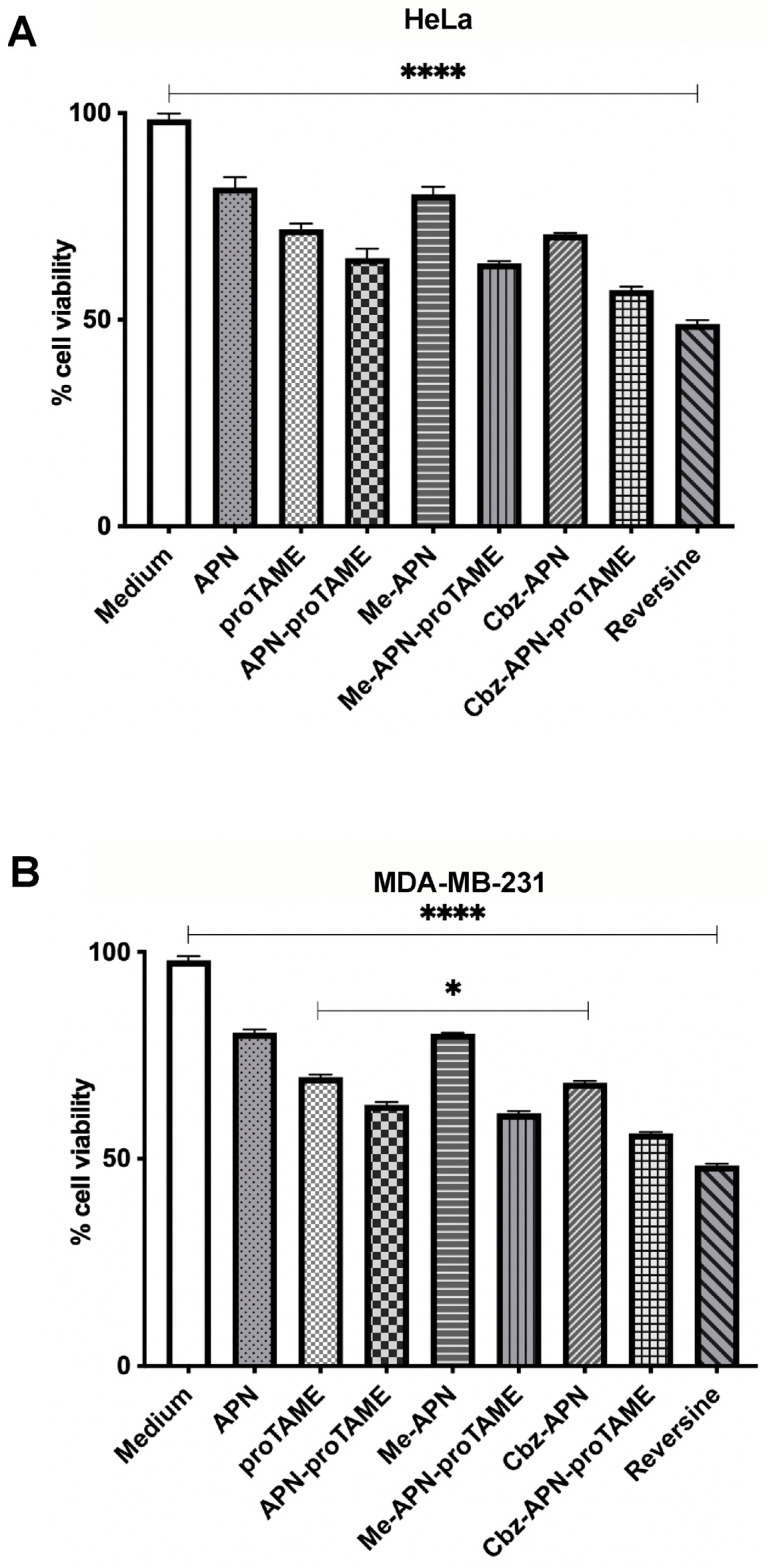
Synergistic effect of APN analogues with proTAME. All APN analogues were tested at 25 μM, and reversine was tested at a 5 μM concentration in (**A**) HeLa and (**B**) MDA-MB-231 cells. The bars represent the mean value of three independent experiments, and the lines represent the standard deviation. One asterisk (*) indicates a *p* value less than 0.05 (*p* ≤ 0.05). Four asterisks (****) indicate a *p* value less than 0.0001 (*p* ≤ 0.0001). Details about the number of cells seeded, temperature, incubation times with the compounds, etc., for these cytotoxicity assays are described in the Materials and Methods.

**Figure 7 biomolecules-14-01439-f007:**
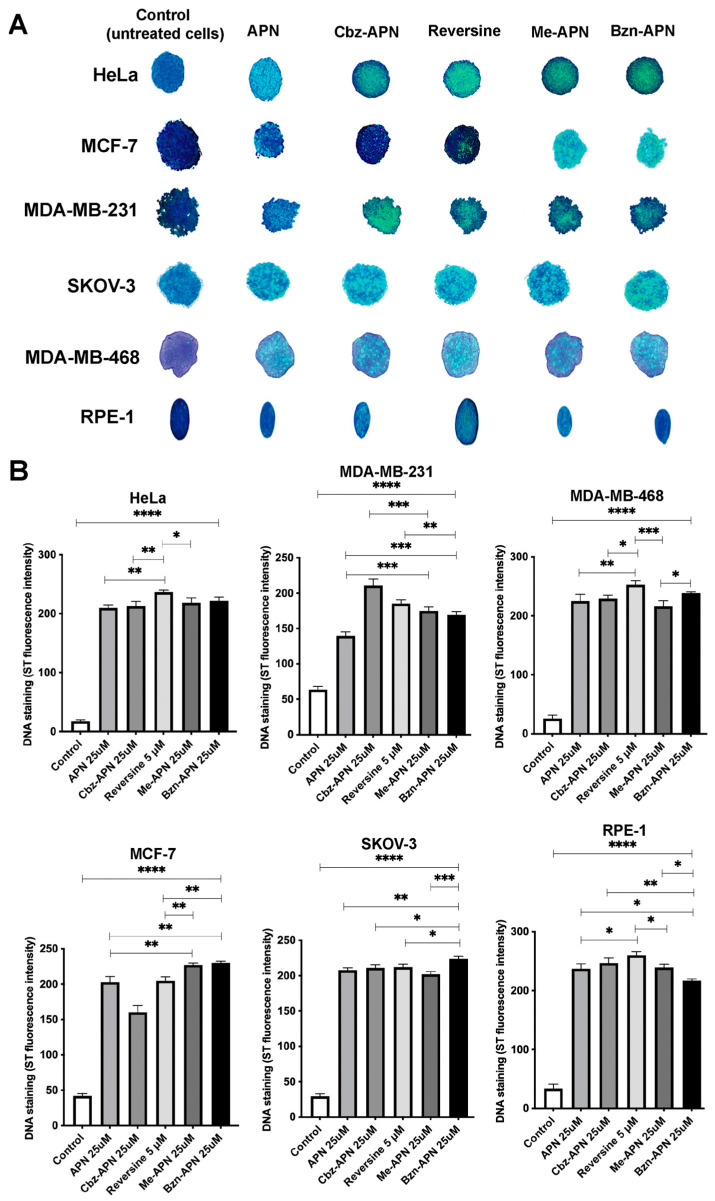
Antimitotic activity of the new APN analogues in 3D cells of different tissue origins. (**A**) APN, Me-APN, Cbz-APN and Bzn-APN were tested in Hela, MCF-7, MDA-MB-231, MDA-MB-468, SKOV-3 and RPE-1 spheroids. (**B**) Quantitative densitometry analysis of the treated spheroids. Three independent experiments were conducted, with similar results observed in all the replicas. Details about the number of cells seeded, temperature, incubation times with the compounds, etc., for the spheroid assays are described in the Materials and Methods. One asterisk (*) indicates a *p* value less than 0.05 (*p* ≤ 0.05), two asterisks (**) indicate a *p* value less than 0.01 (*p* ≤ 0.01), three asterisks (***) indicate a *p* value less than 0.001 (*p* ≤ 0.001), and four asterisks (****) indicate a *p* value less than 0.0001 (*p* ≤ 0.0001).

**Figure 8 biomolecules-14-01439-f008:**
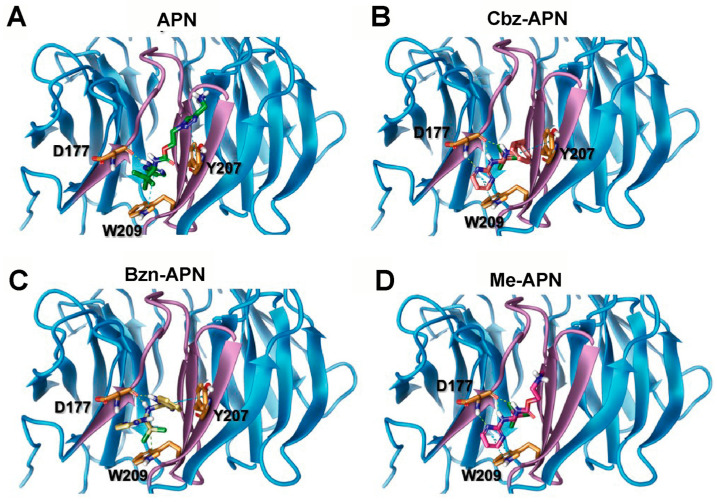
Computational docking of Cdc20–APN analogues. (**A**) The crystal structure of Cdc20 with apcin (the latter is shown in green) was used for computational docking analysis of Cdc20–APN analogues. (**B**) Cbz-APN (red); (**C**) Bzn-APN (yellow); (**D**) Me-APN (pink). Key Cdc20 residues implicated in the interactions are shown in orange. H-bonds are shown in yellow dashed lines and pi–pi interactions in green. APN and analogues are shown in stick representation.

**Table 1 biomolecules-14-01439-t001:** Predicted pharmacological profiles of selected compounds based on QikProp and Glide docking scores.

Compound ID	G Score	QPlogS ^a^	QPlog HERG ^b^	QPP Caco ^c^	QPlog BB ^d^	% Oral Absorption ^e^
APN	−6.12	−5.91	−5.86	185.34	−1.40	75.56
Me-APN	−7.67	−3.93	−6.24	201.28	−0.28	84.76
Bzn-APN	−6.57	−2.95	−5.95	676.79	0.47	94.49
Cbz-APN	−6.33	−6.07	−6.26	1933.22	−0.11	100.00

^a^: Predicted aqueous solubility [−6.5/0.5]; ^b^: HERG K+ channel blockage (log IC50) [concern below −5]; ^c^: Apparent Caco-2 cell permeability in nm/s [<25 poor, >500 excellent]; ^d^: Predicted log of the brain/blood partition coefficient [−3.0/1.2]; ^e^: Human oral absorption in the GI [<25% is poor] [range of 95% of drugs].

**Table 2 biomolecules-14-01439-t002:** Predicted interactions of APN and the new APN analogues with cytochromes P450 (CYP), as calculated with swissADME [28].

Compound ID	CYP Substrate/Inhibition				
	CYP1A2	CYP2C19	CYP2C9	CYP2D6	CYP3A4
APN	NS/NI	S/I	S/I	NS/NI	S/I
Me-APN	NS/NI	S/I	NS/NI	NS/NI	S/I
Bzn-APN	S/I	S/I	NS/NI	S/I	S/I
Cbz-APN	S/I	S/I	S/I	NS/NI	S/I

## Data Availability

The original contributions presented in this study are included in the article/Appendix A. Further inquiries can be directed to the corresponding author/s.

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
