# Peer review of "Targeting APC/C Ubiquitin E3-Ligase Activation with Pyrimidinethylcarbamate Apcin Analogues for the Treatment of Breast Cancer"

_biomolecules, 2024, doi:10.3390/biom14111439_

Round 1
Reviewer 1 Report
Comments and Suggestions for Authors
The work of Kapanidou et al. is quite interesting and timely. Overall, the manuscript is well written and presents an appropriate list of references. However, some minor and major issues must be resolved in order to be allowed its publication. These are listed as follows:
Minor comments
Line 3: By considering the cell lines used in this work, manuscript’s title must be centered not only in breast cancer but in “female cancers”.
Line 79: “The fact that TAME is not…”
Line 156: Info regarding to type, origin and malignity for each cancer cell line, as well as for fibroblasts and RPE-1, must be included into M&M section.
Line 212: SUM-159 cells are only mentioned here, but never used. So, it must be eliminated.
Line 224: Authors stated that statistical analysis was performed. So, figures 3 and 6 must have indicated with an asterisk those comparisons in which the P value was minor or equal to 0.05. Of course, discussion and conclusions must be expressed in function on that statistical significance.
Line 349: Following “… exposure (Figure 3).” insert the following: “The measured cytotoxicity of APN at 25 uM… agreement with previous reports by others [13]” (ie. lines 360-361). Thus, it is better explained why authors used 25 uM APN and analogues in this experiment.
Line 359: “A neglectable cytotoxicity was observed…”
Line 375: There are not titles in Y-axis of graphs for MCF-7 and RPE-1 in Fig 3.
Line 378: Legend in Fig 3 must be self-contained, i.e. understandable without reference to the text. Thus, delete phrase “The MTT assays showed… on fibroblasts at 25 uM concentration”, because it corresponds to a description of results. Instead, please add information of the experiment, for instance, number of cells, compound concentrations, temperature, time, etc. Also, delete all numeric data of compounds (ie. lines 381-391), instead of this please indicate the value (e.g. *p<0.05; **p<0.001) corresponding to each statistical analysis between treatments, as stated in M&M.
Line 402: “In agreement with previous reports [33,34], time-course inhibition of Mps1 kinase with reversine impaired a SAC response, resulting in accelerated mitosis progression, Mps1 proteolytic degradation and ultimately cell death”. However, Mps1 proteolytic degradation was not determined in this experiment. Consider deletion.
Line 407 (Fig 4A): the use of either Qdot 605-A or PE-Texas Red-A is confused. Please provide a rational explanation of this in M&M section and also figure legend.
The paragraph “APN and its analogues Cbz-APN and… MDA-MB-231 compared to untreated cells” must be included into the main text, as it is a description of results. Replace it with a self-contained legend of experimental conditions.
Figure 4B (legend): Use uM instead “micromolar”.
Delete phrase “As anticipated, APN… of Cyclin B and Cdc20” as it corresponds to results.
Delete phrase “Inhibition of the upstream SAC… checkpoint protein kinase” as is unrelated to the figure. In fact, Mps1 proteolytic degradation was not determined here.
Line 435: delete phrase “These experiments confirmed… proliferation of RPE-1 cells at 25 uM concentration” because it’s a description of results. Instead, please include a self-contained, understandable figure legend.
Line 460: “Based on these findings, in this study APN, Me-APN and Cbz-APN were used at 25 uM alone and in combination with 10 uM proTAME and tested in HeLa and MDA-MB-231 cells (Figure 6).”
Line 465: delete “Figure 6A,B shows the results obtained for HeLa and MDA-MB-231 cancer cells only”.
Line 468: “…therapies may be key to enhance a therapeutic response in breast, cervical and ovarian cancer cells (Supplementary Figure X)”. Supp Fig X: results from MCF-7 and SKOV-3 cells.
Line 483: a right title for Fig 6 is: “Synergistic effect of APN analogues with proTAME in MDA-MB-231 and HeLa cells”.
Line 485: The phrase “The Cbz-APN-proTAME combination showed the highest cell proliferation inhibition effect in these cancer cell types” is a description of results, so replace it with a proper self-contained, understandable figure legend in which authors explicit the assay used and conditions (i.e. time, temp, etc).
Line 489: Delete numerical data from line 489 to 495.
Line 499: “Following the observation that APN analogues exhibited the highest cytotoxicity to the female cancer cell lines, but were far less cytotoxic to normal cells in 2D cells in culture, we set…”
Line 504: “…spheroids of Hela, MCF-7, MDA-MB-231, and RPE-1 cells where generated and the cytotoxicity of APN, Me-APN, Cbz-APN and Bzn-APN tested.”
Line 510: “In a nutshell, Cbz-APN proved to be a potent cytotoxic compound against breast and cervical spheroids…”
Line 519: The phrase “These assays showed that compared to untreated cells, APN and the new compounds Me-APN, Cbz-APN and Bzn-APN greatly induced cell death, as based on the amount of DNA stained by the green fluorescence dye SYTOX upon cell lysis” is a description of results, so replace it with a proper figure legend.
Major comments
Figure 4A: This experiment in quite relevant for the main conclusion of the work. So, if authors have shown results with Hela cells treated with 3 compounds, same must be also shown with normal cells (fibroblasts or RPE-1). Reversine treatment must be included in both Hela and normal cells. In addition, results obtained with untreated and treated MCF-7, MDA-MB-231 and normal cells can be provided, for instance, as supplementary figures.
Figure 4B: In concordance with 4A, authors must include both anti-cyclin B and anti-cdc20 blots, but from Hela cells, which were untreated and treated with APN, Bzn-APN, Cbz-APN and reversine. Like in 4A, blots of MCF-7, MDA-MB-231 and normal cells can be provided as supplementary figures.
Figure 5: It’s quite hard to observe inhibition of clone formation as stated by authors. Even an increased number can be observed at low concentration of Cbz-APN. Likewise, it appears that lesser number of either MCF-7 or MDA-MB-231 cells (see untreated) were seeded in plates. Thus, as author have stated that experiments were performed 3 times with similar numbers in all replicas, please include a graph of quantitation of clones together a corresponding statistical analysis (ie. P value) in order to show significant differences.
Figure 6: There are not titles in Y-axis of graphs. Notably, levels of bars appear surprisingly similar in both graphs. It must be confirmed.
It’s suggested the following order is bars: medium - APN - pro - APN/pro - MeAPN - MeAPN/pro - CbzAPN - CbzAPN/pro - reversine. This will allow to properly compare between similar conditions.
Please include the same experiment performed in normal fibroblasts and/or RPE-1 cells as control.
Authors must perform a statistical analysis and include P values (as asterisks) in all graphs.
Put in the main manuscript the results from Hela, MDA-MB-231 and fibroblast/RPE-1 cells, while results from MCF-7 and SKOV-3 cells can be provided as supplementary figures.
Figure 7: It’s quite hard to observe the effect of individual compounds in precluding spheroidogenesis without a quantitative determination of DNA staining. Authors must perform a digital analysis of this and further to construct a graph with numerical data, in which a proper statistical analysis is displayed with P values as asterisks. Perform the above with only Hela, MCF-7, MDA-MB-231, and normal cells as control. Data from MDA-MB-468 and SKOV-3 cells can be provided as a supplementary figure.
Optional: Authors should consider to change the order of displaying results. Thus, after figure 7 (i.e. line 524), display data related to Cbz-APN effect on clonogenic capacity of Hela, MCF-7, MDA-MB-231 and RPE-1 cells (i.e. lines 423 – 440). In this case, paragraph should initiate with “Finally, in order to confirm the highest cytotoxic…”.
Comments on the Quality of English LanguageMinor editing of English language is required.
Reviewer 2 Report
Comments and Suggestions for Authors
The authors provide a detailed description of the potential of the pyrimidinalcarbamate APN analogues Me-APN, Cbz-APN and Bzn-APN 686 in relation to cytotoxic activity against breast cancer cells, according to the experimental tests presented in their manuscript. I consider that it is a good, experimentally very complete manuscript, which can give a start to the study of APN analogues, as the authors mention, not only in breast cancer, but also to generate knowledge in the pharmacological area for the synthesis of anticancer molecules. It is suggested to review the quality of Figs. 4 and 7.
Round 2
Reviewer 1 Report
Comments and Suggestions for Authors
Please, significantly improve quality of figures at the final version.
Author Response
We agree and thank the reviewer for this observation. We will make sure that high quality figures are incorporated in the final version.